# Phase Separation in Chromatin Organization and Human Diseases

**DOI:** 10.3390/ijms26115156

**Published:** 2025-05-28

**Authors:** Ziwei Zhai, Fei Meng, Junqi Kuang, Duanqing Pei

**Affiliations:** 1Centre for Regenerative Medicine and Health, Hong Kong Institute of Science & Innovation, Chinese Academy of Sciences, Hong Kong SAR, China; zwzhai@outlook.com (Z.Z.); fei.meng@crmh-cas.org.hk (F.M.); 2Laboratory of Cell Fate Control, School of Life Sciences, Westlake University, Hangzhou 310024, China; 3Westlake Laboratory of Life Sciences and Biomedicine, Hangzhou 310024, China

**Keywords:** chromatin organization, phase separation, chromatin structural dysregulation, human disease, transcription regulation

## Abstract

Understanding how the genome is organized into multi-level chromatin structures within cells and how these chromatin structures regulate gene transcription influencing animal development and human diseases has long been a major goal in genetics and cell biology. Recent evidence suggests that chromatin structure formation and remodeling is regulated not only by chromatin loop extrusion but also by phase-separated condensates. Here, we discuss recent findings on the mechanisms of chromatin organization mediated by phase separation, with a focus on the roles of phase-separated condensates in chromatin structural dysregulation in human diseases. Indeed, these mechanistic revelations herald promising therapeutic strategies targeting phase-separated condensates—leveraging their intrinsic biophysical susceptibilities to restore chromatin structure dysregulated by aberrant phase separation.

## 1. Introduction

In human cells, approximately two meters of DNA must be compressed and dynamically organized into ordered chromatin structures within the limited nuclear space to ensure proper genome replication and the spatiotemporal expression of over 20,000 genes [1,2]. Advances in microscopy and chromosome conformation capture technologies have revealed that genomic DNA is hierarchically organized into nucleosome, chromatin fiber, chromatin loop, topologically associating domain, chromatin compartment, and chromosome territory (Figure 1) [3,4,5,6,7]. Since the formation and remodeling of chromatin structures at each level are closely linked to cell differentiation and embryonic development, their dysregulation often leads to the onset and progression of human diseases, such as developmental disorders and cancers [8,9].

Currently, the primary mechanisms driving chromatin structure formation and remodeling are chromatin loop extrusion and phase separation-mediated chromatin compartmentalization [10,11,12]. Phase separation, a fundamental physicochemical concept, refers to the demixing of two mixed substances with distinct physicochemical properties under force, a phenomenon ubiquitous in nature [13]. In cells, phase separation involves biomacromolecules (proteins, nucleic acids, polysaccharides, lipids, and so on) forming membraneless organelles and subcellular structures via weak transient and multivalent interactions, such as P granules [14], stress granules [15], and nucleoli [16], which regulate diverse biochemical reactions and are often associated with intrinsically disordered regions (IDRs) of proteins [17]. Chromatin compartmentalization similarly relies on these weak transient and multivalent interactions, proceeding through processes such as liquid-liquid phase separation (LLPS), polymer-polymer phase separation (PPPS; also known as bridging-induced phase separation (BIPS)), liquid-gel phase separation (LGPS) or phase separation coupled to percolation (PSCP), and liquid-solid phase separation (LSPS), ultimately organizing chromatin into functional domains to regulate transcription [18,19,20].

In this review, we explore the relationship between chromatin structure and phase separation and describe how phase-separated condensates organize chromatin structure at multiple levels and regulate gene transcription. Furthermore, we discuss recent discoveries in phase separation-mediated chromatin structural dysregulation in cancers and developmental disorders. Finally, we summarize these insights and highlight key unresolved challenges and future perspectives.

## 2. Organization of Multi-Level Chromatin Structure via Phase Separation

### 2.1. Nucleosome and Chromatin Fiber

The nucleosome is the fundamental unit of chromatin. A single nucleosome or nucleosome core particle consists of a histone octamer wrapped by 146 base pairs (bp) of DNA [21]. Together with 46 bp linker DNA and linker histone H1, nucleosomes form a nucleosome array or “beads on a string”, which further folds into a chromatin fiber (Figure 1A) [22]. As early as the 1980s, cryo-electron microscopy revealed liquid-like behavior of chromatin fibers [23]. Subsequent studies demonstrated that nucleosome arrays spontaneously condense into such liquid-like chromatin fibers [24,25], which may be a key feature enabling the genome to manifest self-organized criticality (SOC; a complex systems theory that describes self-organization and emergent order in non-equilibrium systems) gene expression control to determine cell fate [26,27]. Consequently, the hypothesis that nucleosomes or their components drive chromatin compaction and fiber self-assembly via liquid-liquid phase separation (LLPS) was proposed.

The above hypothesis is increasingly supported by recent evidence. The N-terminal tail domains of core histones H2A/H2B and H3/H4 regulate nucleosome core particles (NCPs) formation and oligomerization [28]. NCPs undergo phase separation under physiologically mimetic conditions and the tail domains of H3/H4 independently drive LLPS with DNA [29]. Deletions or mutations of basic residues (K16, R17, R19, K20 and L22) in these tails abolish condensate formation [29,30]. Additionally, H2A can undergo LLPS with DNA, forming condensates that recruit other histones and promote NCPs assembly. H2B, which dimerizes with H2A, tends to deposit under similar conditions [31]. The acidic residues mutations (E61, E64, D90, and E92) do not impede NCPs condensation [30]. However, H2B monoubiquitination, a modification critical for nucleosome oligomerization and transcription, is regulated by the phase separation of transcriptional regulatory protein LGE1 [32]. Computational modeling further suggests that epigenetic modifications, including DNA methylation, histone post-translational modifications, etc., enhance nucleosome plasticity, enabling multivalent interactions essential for LLPS and chromatin fiber stability (Figure 1A) [33].

Beyond core histones, linker DNA and linker histone H1 contribute to phase separation-mediated chromatin structure formation and maintenance. The C-terminal tail of H1, an intrinsically disordered region, forms gel-like condensates with single-stranded DNA [34]. These condensates accumulate during DNA damage, co-localizing with proliferating cell nuclear antigen (PCNA) to stabilize stalled replication forks [34]. H1 also undergoes LLPS with double-stranded DNA. Phosphorylation of its serine residues (S157, S175, and S193) does not prevent condensate formation but alters their internal architecture and promotes H1/DNA dissociation [35]. Notably, H1 preferentially undergoes LLPS with nucleosome array containing 10n + 5 bp linker DNA. Under identical physiological salt concentrations, NCPs require ~200 micromole to initiate phase separation, whereas 12-mer nucleosome arrays with linker DNA and H1 form liquid condensates at ~100 nanomole [30]. These findings collectively demonstrate that linker DNA and histone H1, like core histones, regulate chromatin fiber organization via phase separation (Figure 1A).

### 2.2. Chromatin Loop and Topologically Associating Domain

Following folding into chromatin fibers, nucleosome arrays are extruded into chromatin loops by ATP-dependent structural maintenance of chromosomes complexes (SMCs) [36]. Loop extrusion begins with the recruitment of SMCs components (cohesin, SMC5-SMC6, condensin and so on) by loading factors such as NIPBL and MAU2 [37,38]. The process halts when SMCs encounter a pair of convergently oriented CCCTC-binding factors (CTCF) (Figure 1B) [39,40] and dissociate from chromatin via the chromatin-releasing factor WAPL [41,42]. Through iterative action of these loop-extruding factors, chromatin loops cluster to form topologically associating domains (TADs) (Figure 1B) [10,43]. High-resolution live-cell imaging and high-throughput chromosome conformation capture (Hi-C) analyses reveal that TADs are chromatin structures spanning 0.2–1 Mb, characterized by significantly higher intra-domain DNA interaction frequencies compared to inter-domain regions [5,6,44,45,46]. The roles of SMCs, scanning the genome and bridging distal DNA locus, enable chromatin loops and TADs are to participate in critical biological processes such as promoter-enhancer interactions (Figure 1B) [47,48,49], DNA double-strand break repair [50,51,52], and antibody diversification [53,54,55].

Recent studies highlight phase separation as a critical mechanism for the formation, maintenance, and functional regulation of chromatin loops and TADs. Disrupting phase separated condensates with low-concentration 1,6-hexanediol significantly reduces long-range chromatin interactions and destabilizes ~20% of TADs [56,57,58]. In mammalian cells, coactivators BRD4 and MED1 form phase-separated condensates at super-enhancers locus, scaffolding transcriptionally active chromatin loops and TADs by recruiting transcription factors, RNA polymerase II (Pol II), and other coactivators, thereby enhancing promoter-enhancer interactions (Figure 1B) [59,60,61,62,63]. Subsequently, phosphorylation of C-terminal domain by cyclin-dependent kinases 7/9 (CDK7/9) triggers Pol II release from super-enhancers and incorporation into splicing-associated condensates, promoting efficient transcription of lineage-specific genes [64,65]. For example, the phase-separated condensates containing OCT4, a core pluripotency factor, not only participate in super-enhancer formation to sustain the self-renewal of stem cells but also establish pluripotency-associated chromatin loops and reorganize TADs to promote somatic cell reprogramming [66]. In B lymphocytes, intriguingly, gel-like condensates between spatially remote immunoglobulin heavy-chain locus facilitate chromatin looping for antibody diversification [67].

Phase separation may also regulate transcriptionally repressive chromatin loops and TADs. Hi-C studies show that polycomb repressive complexes (PRC1/2) form transcriptionally repressive loops at enhancer-promoter or promoter-promoter regions [68,69,70,71], which further compact into polycomb-associated domains (PADs) to silence genes during development in mouse and drosophila [72,73,74,75]. PRC1 subunits CBX2 and PHC1 undergo phase separation in vitro and in vivo. In detail, CBX2 drives PRC1 recruitment and condensation in specific chromatin sites, while PHC1 stabilizes condensates to enhance H2A ubiquitination [76,77,78,79,80]. A proposed model suggests that PRC2 binds DNA-methylated regions, deposits H3K27me3, then recruits PRC1 to assemble tightly phase-separated PADs that exclude transcription activators and silence gene expression [81].

In addition to phase-separated proteins, noncoding RNAs (ncRNAs) are emerging as key regulators of chromatin loops and TADs. ncRNAs or ncRNA-DNA hybrids (R-loops) recruit structural proteins CTCF and cohesin to stabilize chromatin loops and TADs boundaries (Figure 1B) [82,83,84,85]. Furthermore, transcriptionally repressive chromatin loops and TADs harbor ncRNAs that form phase-separated condensates with RNA-binding proteins for gene silencing by recruiting SMRT/HDAC1-associated repressive complexes [86,87]. Notably, advanced RNA-DNA interaction mapping techniques and single-molecule fluorescence in situ hybridization have identified abundant ncRNAs within canonical chromatin loops and TADs [87], such as Eleanors in estrogen receptor-associated TADs [88]. Given the ability of ncRNA to drive phase separation and regulate transcription [89,90,91,92], ncRNA-mediated phase separation may represent a novel mechanism for chromatin loops and TADs regulation.

### 2.3. Chromatin Compartment and Chromosome Territory

Individual TADs have a propensity to self-assemble into higher-order chromatin compartments (Figure 1C) [2,93], corresponding to fluctuations of epigenetics and transcription during cell fate transition [26,27,94,95]. Hi-C and multiplex fluorescence in situ hybridization analyses reveal that chromatin compartments typically span 3–5 megabases and are classified into A compartments and B compartments [3,5]. A compartments are associated with transcriptionally active euchromatin, predominantly localized in the nuclear interior and around nuclear speckles, while B compartments correlate with transcriptionally repressive heterochromatin, frequently positioned within perinuclear lamina and around the nucleolus. Consequently, interactions between compartments of the same type occur more frequently than between different types [96,97,98]. In addition, chromosome territories represent chromosome-specific nuclear regions, typically 2–3 μm in diameter (Figure 1C). Chromatin compartments can interact both within the same chromosome territory and across distinct territories [4,99,100].

The formation of chromatin compartments is driven by weak transient and multivalent interactions between chromatin locus, which are intrinsically linked to phase separation [101,102]. Heterochromatin is the first chromatin compartment that has been proven to be formed by phase separation [11,103]. Specifically, phase-separated heterochromatin protein 1 (HP1) recognizes H3K9 methylation, a hallmark of constitutive heterochromatin, and further compacts heterochromatin by altering nucleosome conformation, promoting heterochromatin compartmentalization and gene silencing [104,105]. Conversely, histone acetylation (H3K27ac, H4K16ac, and so on) catalyzed by histone lysine acetyltransferases CBP/p300 marks transcriptionally active euchromatin and defines A compartments in mammalian cells [5,96,106,107]. Acetylated chromatin forms phase-separated condensates through interactions with histone acetylation-binding proteins such as BRD4 and MED1 [30]. Similar phase-separated condensates driven by CTCF can enhance interactions between A compartments [108]. Intriguingly, histone acetylation impedes linker histone H1-mediated chromatin condensation in the absence of acetylation-binding proteins. Furthermore, acetylated chromatin-BRD4/MED1 condensates are mutually exclusive with H1-mediated chromatin condensates [30]. Given that H1 compacts heterochromatin via phase separation [109] and coactivators MED1/BRD4 form condensates on acetylated chromatin while excluding heterochromatin [110], phase separation not only drives chromatin compartments formation but also enforces their sequestration (Figure 1C). For example, active chromatin marks drive spatial sequestration of heterochromatin in Caenorhabditis elegans nuclei [111,112]. Moreover, BAZ2A, a nuclear body component, forms phase-separated condensates with active chromatin while stabilizing H3K27-methylated heterochromatin in mouse embryonic stem cells [113].

Under certain conditions, heterochromatin and euchromatin form specialized chromatin compartments with distinct subcellular structures (Figure 1C) [110,114]. Prominent examples include lamina-associated domains (LADs), where heterochromatin interacts with the nuclear lamina [97,115], and nucleolus-associated domains (NADs) surrounding the nucleolus [96,116,117]. Lamin-B receptor regulates heterochromatin compaction by binding HP1 and modulating its phase separation [103], while lamina-heterochromatin interactions maintain B compartments positioning and nuclear architecture, including pericentric and telomeric heterochromatin [12,98,118]. The nucleolus, a canonical phase-separated membraneless organelle [119,120], relies on RNA helicase DDX18 and nucleophosmin NPM1 to stabilize NADs and enforce gene silencing [121]. Differently, transcriptionally active euchromatin associates with nuclear speckles to form nuclear speckle-associated domains (SPADs) [122,123], where scaffold proteins SON and SRRM2 drive speckle assembly via phase separation [124,125,126]. SPADs overlap > 95% with A compartments. Depleting the loop-extruding factor NIPBL disrupts SPADs formation and gene expression [127]. Interestingly, under persistent DNA damage, serine-protein kinase ATM collaborates with 53BP1-phosphorylated H2AX (γH2AX), a marker of DNA double-strand breaks [128], to form phase-separated chromatin compartments (D compartments) detectable by Hi-C, promoting the activation of the DNA damage response and repair [129,130].

Similarly, chromatin territories are also regulated by phase separation (Figure 1C) [101,102]. X-chromosome inactivation in female mammals, which ensures dosage compensation during development [131], involves the long noncoding RNA Xist localizing to one X chromosome and recruiting RNA-binding proteins (PTBP1, SPEN, HNRNPK) via its E-repeat sequence. This process induces phase separation, condensing the X chromosome into a Barr body (X-chromosome territory) for gene silencing [132,133,134]. In addition, chromatin undergoes phase separation to form tightly compacted, negatively charged territories through condensin and deacetylase during mitosis. These territories repel negatively charged spindle microtubules, preventing microtubules perforation and ensuring genome segregation [135].

## 3. Dysregulation of Chromatin Structure Mediated by Phase Separation in Human Diseases

### 3.1. Cancers

As mentioned above, chromatin organization is broadly regulated by phase separation. Aberrant phase separation caused by genetic mutations or pathological conditions frequently disrupts physiological chromatin structure, directly contributing to various cancers (Table 1) [52,101,136,137,138]. Notably, 16.5% of cancers harbor chromosomal translocations that generate oncogenic fusion proteins [139]. A major class of these fusion proteins combines intrinsically disordered regions (IDRs) with chromatin-interacting domains [140,141], underscoring phase separation-driven chromatin structural dysregulation as a widespread pathological mechanism [142,143]. For instance, in multiple acute leukemia subtypes, the IDR of nuclear pore complex protein NUP98 fuses with transcription factors or chromatin-interacting factors [144,145]. The NUP98-HOXA9 fusion protein forms phase-separated transcriptionally active chromatin loops at proto-oncogenic locus, further assembling super-enhancers to amplify oncogene activation [146]. Moreover, NUP98 fusions with KDM5A, LNP1, PRRX1, or NSD1 likely exert comparable effects [147,148,149,150]. Similar mechanisms are observed in FET family (FUS, EWS, TAF15) fusions [151], BRD4-NUT fusions [152], and YAP fusions [153], which form oncogenic condensates that recruit ATP-dependent chromatin remodeling complexes BAFs (Brg/Brahma-associated factors), histone acetyltransferase p300, and Pol II to enhance chromatin accessibility and oncogene expression, driving malignancies such as sarcomas [154,155,156], midline carcinomas [157,158,159], and ependymomas [153].

Notably, the epigenetic regulator SS18 fuses with SSX1 in synovial sarcoma, forming the oncogenic SS18-SSX1 fusion (Table 1). This fusion protein forms condensates at H2AK119ub-marked oncogenic locus, recruiting the BAFs and histone acetyltransferase CBP/p300 to assemble transcriptionally active chromatin loops/TADs. Simultaneously, it strongly excludes HDAC1/2 deacetylase complexes, elevating H3K27ac levels and sustaining oncogene overexpression [160,161,162]. Remarkably, subunits of the BAFs, which mobilize nucleosomes to increase chromatin accessibility, are mutated in 19.6% of human cancers [163,164]. Furthermore, BAFs assembly and function are regulated by phase separation of its subunits [165,166,167,168,169], implicating phase separation-mediated chromatin remodeling as a key mechanism in tumorigenesis.

Cancer-associated mutations not only induce pathological phase separation but also disrupt physiological condensate dynamics, destabilizing chromatin structure (Table 1). A prime example is the histone demethylase UTX, a pan-tumor suppressor whose missense mutations drive pancreatic cancer and myeloid leukemia [170,171,172]. UTX forms phase-separated condensates in vitro and in vivo, recruiting histone lysine methyltransferase MLL4 and p300 to establish transcriptionally active chromatin loops that activate immune-related genes while suppressing cell division-related genes [173]. However, tumor-associated mutations in UTX-IDR impair its phase separation, destabilizing chromatin loops and promoting carcinogenesis [173]. Similarly, the pioneer transcription factor FOXA1, which forms anti-heterochromatin condensates and activates tumor suppressor genes, is frequently mutated in breast and prostate cancers. Mutations in DNA-binding domain of FOXA1 abrogate its tumor-suppressive function driven by heterochromatin targeting and condensate formation [174,175,176,177,178,179].

Aside from genetic mutations, unique tumor microenvironments drive aberrant phase separation and chromatin structural dysregulation (Table 1). For instance, hypoxia, a hallmark of aggressive solid tumors [180,181], induces transcription factor ZHX2 to form phase-separated condensates that recruit CTCF, BRD4, and MED1, reshaping chromatin loops to activate oncogenes and promote metastasis [182]. Similarly, hyperactivated ARID1A, phospho-HDAC6 and FOXM1 in Ewing’s sarcoma and breast cancer remodel chromatin structure via phase separation, driving oncogenic transcription and tumor progression [168,183,184]. Furthermore, telomerase-negative cancers employ homology-directed repair to elongate telomeres and stabilize telomeric heterochromatin for immortalization [185]. In these cells, overexpressed telomeric repeat-containing RNA (TERRA) collaborates with histone lysine demethylase LSD1 and RNA-binding protein HNRNPA1 to form phase-separated telomeric condensates. These condensates stabilize R-loops and promote telomeric capping, maintaining telomeric heterochromatin integrity [186,187,188].

**Table 1 ijms-26-05156-t001:** Key genes involved in the dysregulation of chromatin structure mediated by phase separation in cancers.

Genes	Pathological Functions Through Chromatin Organization Associated with Phase Separation	Refs.
NUP98 fusions with HOXA9, KDM5A, LNP1, PRRX1, NSD1	NUP98 fusion proteins form phase-separated condensates and promote transcriptionally active chromatin loops at proto-oncogenic locus, further assembling super-enhancers to amplify oncogene activation in human hematological malignancies.	[146,147,148,149,150]
FET family fusions (FUS, EWS, TAF15)	FET family fusion proteins condensate at specifically silenced locus and attract RNA polymerase II and chromatin remodeling complexes BAFs to form transcriptionally active chromatin hubs, contributing to oncogenic transformation in sarcomas and leukemia.	[154,155,156]
YAP-MAMLD1, C11ORF95-YAP	YAP fusion proteins nuclear condensates concentrate transcription factors and coactivators (TEAD, BRD4, MED1) and exclude polycomb repressive complex PRC2, inducing transcriptionally active chromatin loops that promote ependymoma tumorigenesis.	[153]
BRD4-NUT	BRD4-NUT recognizes acetylated chromatin and binds acetyltransferase p300 to form condensates, inducing histone hyperacetylation and chromatin subcompartment that sustain aberrant anti-differentiation genes transcription and perpetual tumor cell growth in midline carcinomas.	[157,158,159]
SS18-SSX1	SS18-SSX1 condensates at H2AK119ub-marked oncogenic locus, recruits BAFs complexes and histone acetyltransferase CBP/p300 while excludes HDAC1/2 deacetylase complexes to assemble transcriptionally active chromatin loops/TADs that elevate H3K27ac level and sustain oncogene overexpression in synovial sarcoma.	[160,161,162]
UTX	UTX condensates demethylate H3K27me3 and recruit histone lysine methyltransferase MLL4 and p300 to establish transcriptionally active chromatin loops that activate immune-related genes while suppressing cell division-related genes. The pancreatic cancer and myeloid leukemia-associated mutations in UTX impair these condensates.	[173]
FOXA1	FOXA1 condensates unpack heterochromatin and activate tumor suppressor genes. Mutations in DNA-binding domain of FOXA1 abrogate its tumor-suppressive function driven by heterochromatin targeting and condensate formation in breast and prostate cancers.	[174,175,176,177,178,179]
ZHX2	ZHX2 condensates, in response to hypoxic tumor microenvironment, recruit CTCF, BRD4, and MED1, reshaping chromatin loops to activate oncogenes and promote metastasis	[182]
ARID1A, HDAC6, FOXM1	Hyperactivated ARID1A/phospho-HDAC6/FOXM1 forms similar condensates in Ewing’s sarcoma or breast cancer recruiting BAFs complexes, Pol II, and coactivators to remodel chromatin structure that drive oncogenic transcription and tumor progression.	[168,183,184]
TERRA	TERRA, an overexpressed lncRNA in telomerase-negative cancers, collaborates with histone lysine demethylase LSD1 and RNA-binding protein HNRNPA1 to form telomeric condensates which elongate telomeres and stabilize telomeric heterochromatin for immortalization.	[186,187,188]

### 3.2. Developmental Disorders

Phase separation-mediated chromatin structural dysregulation also contributes to developmental disorders, including neurodevelopmental diseases such as Rett syndrome, Kabuki syndrome, and autism (Table 2) [189,190,191,192,193]. Rett syndrome primarily affects females and is clinically characterized by intellectual disability, loss of language function, stereotypic hand movements, and gait abnormalities [194,195], which is strongly linked to mutations in the X-chromosomal methyl-CpG-binding protein 2 (MECP2) [196,197,198]. Recent studies from three independent groups demonstrated that MECP2 undergoes phase separation in vitro and in vivo. MECP2 recognizes and binds methylated DNA, condensing chromatin fibers to form heterochromatin compartments. Rett syndrome-associated MECP2 mutations severely impair its phase separation, leading to related heterochromatin dysregulation. Furthermore, coactivator BRD4 exhibits enhanced binding to MECP2-silenced chromatin, activating related genes, ultimately driving Rett syndrome [189,190,191,199]. In Kabuki syndrome, mutations in histone lysine methyltransferase MLL4 and demethylase UTX reduce their ability to form phase-separated condensates, impairing recruitment of MED1/BRD4 and exclusion of PRC1/2. This disrupts the balance between chromatin compartments, resulting in craniofacial anomalies, postnatal growth retardation, intellectual disability, and organ malformations [173,192,200,201]. Notably, genomic analyses of neurodevelopmental disorders in the mammalian genome reveal 1204 missense and frameshift mutations in chromatin remodeling complexes BAFs, 58.3% of which are neurodevelopmental disorder-specific [193,202]. Overall, given the role of phase separation in regulating BAFs [167], phase separation-mediated chromatin structural dysregulation broadly promotes neurodevelopmental pathogenesis.

Beyond neurodevelopmental disorders, limb malformations caused by severe skeletal defects are also regulated by phase separation-mediated chromatin remodeling (Table 2) [203,204]. For example, in congenital synpolydactyly, HOXD13 mutants with expanded polyalanine form phase-separated condensates that fail to recruit coactivators. This disrupts the formation and maintenance of transcriptionally active TADs, reducing related gene expression and causing synpolydactyly [203,205]. Furthermore, hand-foot genital syndrome (caused by HOXA13 mutations [206]) and cleidocranial dysplasia (caused by RUNX2 mutations [207]) may share similar mechanisms [203]. Additionally, in brachyphalangy, polydactyly, and tibial aplasia/hypoplasia syndrome (BPTAS), HMGB1 undergoes spontaneous mutations that replace its C-terminal IDR with an arginine-rich basic tail. Unlike the wild-type HMGB1 that regulates the organization of chromatin loops, HMGB1 mutant forms aberrant condensates that invade the nucleolus, disrupting its function and leading to BPTAS [204,208]. Remarkably, over 10,000 disease-associated protein C-terminal IDRs harbor similar frameshift mutations [204,209], suggesting that phase-separated dysregulation linked to chromatin organization represents a common pathogenic mechanism.

**Table 2 ijms-26-05156-t002:** Key genes involved in the dysregulation of chromatin structure mediated by phase separation in developmental disorders.

Genes	Pathological Functions Through Chromatin Organization Associated with Phase Separation	Refs.
MECP2	MECP2 recognizes and binds methylated DNA, condensing chromatin fibers to form heterochromatin compartments. Rett syndrome-associated MECP2 mutations impair these compartments, leading to related heterochromatin dysregulation and pathogenic genes activation.	[189,190,191]
MLL4	MLL4 condensates methylate H3K4 and recruit MED1/BRD4 while excluding PRC1/2. Kabuki syndrome-associated mutations in MLL4 impair these condensates and disrupt the balance between chromatin compartments, resulting in transcriptional dysregulation.	[192]
HOXD13, HOXA13, RUNX2	HOXD13/HOXA13/RUNX2 condensates with expanded polyalanine mutation fail to recruit coactivators and disrupt the formation of transcriptionally active TADs, causing synpolydactyly, hand-foot genital syndrome, or cleidocranial dysplasia.	[203]
HMGB1	HMGB1 with arginine-rich basic tail mutation fails to organize the chromatin loops and forms aberrant condensates that invade the nucleolus, disrupting its function and leading to brachyphalangy, polydactyly, and tibial aplasia/hypoplasia syndrome.	[204,208]

## 4. Conclusions

Emerging evidence suggests that numerous phase-separated condensates, recruiting or excluding effectors through their own physicochemical property, underly the organization of multi-level chromatin structure (Figure 1). A series of chromatin structural dysregulations mediated by phase separation are linked to human diseases (Table 1). However, the molecular mechanisms by which phase separation organize chromatin structures, and how phase separation itself is precisely regulated, remain poorly understood and highly debated [20,210,211]. For instance, heterochromatin formation and compaction in living cells may not solely rely on phase separation but could also involve low-valency interactions with spatially clustered binding sites (ICBS) [212,213]. Additionally, endogenous gene transcription may occur independently of phase separation, and condensate formation by transcription factors and coactivators might even suppress transcriptional activity [214,215]. Nonetheless, treatment with 1,6-hexanediol, the only widely used tool to globally disrupt condensates, partially destabilizes chromatin structures in cell [56,57,58], indicating that chromatin organization depends, at least partially, on phase separation. A comprehensive understanding of these processes will require advanced technologies for probing and engineering phase separation to unravel bona fide regulatory mechanisms.

It is worth noting that growing body of studies have explored chromatin structural dysregulation caused by aberrant phase separation in cancers [161,162,183,184,216,217]. The small molecules and peptides used or developed in these studies, targeting phase-separated condensates—leveraging their intrinsic biophysical susceptibilities to restore chromatin structure dysregulated by aberrant phase separation—can help us further explore the molecular mechanisms of chromatin organization and phase separation. Furthermore, such tools may also provide precursors for therapeutic drug development targeting chromatin structural dysregulation-driven and phase separation-driven diseases.

## Figures and Tables

**Figure 1 ijms-26-05156-f001:**
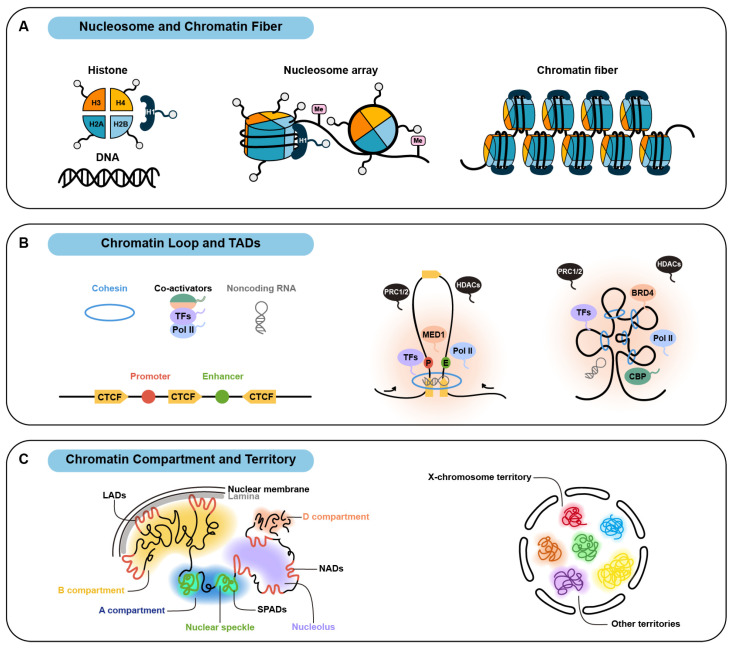
Organization of multi-level chromatin structure via phase separation. (**A**) The phase-separated C-terminal domains or intrinsically disordered regions (IDRs) of histones trigger nucleosomal self-assembly and further organize nucleosome array into liquid-like chromatin fiber. Various DNA and epigenetic modifications, including DNA methylation and histone post-translational modifications impact multivalent interactions essential for phase separation and chromatin fiber stability; (**B**) Chromatin loop (middle) and topologically associating domains (TADs, right), including a transcriptionally active promoter-enhancer loop, organize via loop extrusion mediated by cohesion-CCCTC-binding factor (CTCF) and via phase-separated condensates driven by transcription factors (TFs), coactivators (e.g., mediator complex subunit 1 (MED1) and bromodomain-containing protein 4 (BRD4)), and RNA polymerase II (Pol II) while excluding repressors with IDRs (e.g., polycomb repressive complexes (PRC1/2) and histone deacetylase (HDACs)), where noncoding RNAs (ncRNAs) stabilize structural boundaries; (**C**) Chromatin compartments (left) and chromosome territories (right) form through phase-separated condensates generated by related proteins and ncRNAs. A compartment roughly corresponds to transcriptionally active euchromatin, such as nuclear speckle-associated domains (SPADs). B compartment is generally related to transcriptionally repressive heterochromatin, including lamina-associated domains (LADs) and nucleolus-associated domains (NADs). D compartments are induced by DNA double-strand breaks, contributing to the activation of the DNA damage response and repair. Terminal tails represent phase-separated IDRs. Backgrounds represent phase-separated condensates.

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
