# Peer review of "Phase Separation in Chromatin Organization and Human Diseases"

_ijms, 2025, doi:10.3390/ijms26115156_

Round 1
Reviewer 1 Report
Comments and Suggestions for Authors
The review article "Phase Separation in Chromatin Organization and Human Diseases" by Ziwei Zhai et al. discusses recent findings on the mechanisms of chromatin organization mediated by phase separation focussing on the roles of phase-separated condensates in chromatin structural dysregulation in human diseases. The article is well written and summarizes a huge amount of relevant literature. Nevertheless I would also draw the attention on the following articles which may be included where appropriate (especially in the context of selforganization principles as driving forces):
DOI: 10.4415/ANN_24_01_11
doi.org/10.1371/journal.pone.0167912
doi.org/10.1371/journal.pbio.2000640
doi.org/10.3390/ijms21010240
doi: 10.3390/ijms21134581
doi.org/10.1016/j.ygeno.2021.11.027
doi.org/10.1016/j.plrev.2021.03.001
dx.doi.org/10.1016/j.physa.2016.12.016
In line 77: the 30 nm fiber is controversely discussed. It is not clear whether it really exist. Please mention.
Reviewer 2 Report
Comments and Suggestions for Authors
In this manuscript, Ziwei Zhai and colleagues discuss recent findings on the mechanisms of chromatin organization mediated by phase separation, with a focus on the roles of phase-separated condensates in chromatin structural dysregulation in human diseases. I have following comments:
1, For the Abstract, practical interest of this study should be provided.
2, For the Figure 1, abbreviations appeared in the figure like MED1,BRD4 should be explained in the legend.
3, Key genes regulating phase separation associated with chromatin organization and human diseases should be summarized in Tables.
4, A conclusion section should be included.
Round 2
Reviewer 2 Report
Comments and Suggestions for Authors
Authors have positively respond to my questions in the revision.